# Identification and Characterization of Antibiotic-Resistant, Gram-Negative Bacteria Isolated from Korean Fresh Produce and Agricultural Environment

**DOI:** 10.3390/microorganisms11051241

**Published:** 2023-05-08

**Authors:** Sunyoung Jeong, Ile Kim, Bo-Eun Kim, Myeong-In Jeong, Kwang-Kyo Oh, Gyu-Sung Cho, Charles M. A. P. Franz

**Affiliations:** 1Department of Microbiology and Biotechnology, Max Rubner-Institut, Federal Research Institute of Nutrition and Food, Hermann-Weigmann-Straße 1, 24103 Kiel, Germany; sunyoung.jeong@mri.bund.de (S.J.); ile.kim@mri.bund.de (I.K.); charles.franz@mri.bund.de (C.M.A.P.F.); 2College of Life Sciences and Biotechnology, Korea University, Anam-ro, Seongbuk-gu, Seoul 02841, Republic of Korea; 3Department of Life Science, Handong Global University, 558 Handong-ro, Buk-gu, Pohang 37554, Republic of Korea; 4Microbial Safety Division, National Institute of Agricultural Sciences, Rural Development Administration, 166 Nongsaengmyeong-ro, Iseo-myeon, Wanju 55365, Republic of Korea; kimboeun0415@korea.kr (B.-E.K.); mijeong829@korea.kr (M.-I.J.);

**Keywords:** antibiotic resistance, lettuce, fresh produce, whole genome sequencing

## Abstract

The consumption of fresh produce and fruits has increased over the last few years as a result of increasing consumer awareness of healthy lifestyles. Several studies have shown that fresh produces and fruits could be potential sources of human pathogens and antibiotic-resistant bacteria. In this study, 248 strains were isolated from lettuce and surrounding soil samples, and 202 single isolates selected by the random amplified polymorphic DNA (RAPD) fingerprinting method were further characterized. From 202 strains, 184 (91.2%) could be identified based on 16S rRNA gene sequencing, while 18 isolates (8.9%) could not be unequivocally identified. A total of 133 (69.3%) and 105 (54.7%) strains showed a resistance phenotype to ampicillin and cefoxitin, respectively, while resistance to gentamicin, tobramycin, ciprofloxacin, and tetracycline occurred only at low incidences. A closer investigation of selected strains by whole genome sequencing showed that seven of the fifteen sequenced strains did not possess any genes related to acquired antibiotic resistance. In addition, only one strain possessed potentially transferable antibiotic resistance genes together with plasmid-related sequences. Therefore, this study indicates that there is a low possibility of transferring antibiotic resistance by potential pathogenic enterobacteria via fresh produce in Korea. However, with regards to public health and consumer safety, fresh produce should nevertheless be continuously monitored to detect the occurrence of foodborne pathogens and to hinder the transfer of antibiotic resistance genes potentially present in these bacteria.

## 1. Introduction

Fresh produce and fruits are known to harbor naturally occurring microorganisms with high diversity belonging to a great variety of microbial lineages [1,2]. Their consumption has increased over the last years and they are mostly eaten raw, which increases the risk of exposure to human pathogens such as *Salmonella*, *Listeria monocytogenes*, and Shiga-toxin-producing *Escherichia* (*E.*) *coli*, and/or opportunistic pathogens such as *Klebsiella*, *Enterobacter*, and *Citrobacter* [3,4,5,6]. Regardless, the World Health Organization (WHO) recommends an intake of 400 g of fresh fruits and vegetables on a daily basis based on their effects in lowering the risk of various diseases [7]. Furthermore, as salads have become popular as foods associated with a healthy lifestyle, sales of fresh produce continue to grow. Though it is welcomed that more people enjoy vegetables, this trend raises the necessity of thorough investigation of the quality and safety of fresh produce.

However, the WHO and the Food and Agriculture Organization (FAO), as well as various studies, have identified and shown fresh produce and fruits to be potential sources of human pathogens [3,5,8,9,10]. According to the Centers for Disease Control and Prevention (CDC), from 2010 to 2017, approximately 12.7% of foodborne outbreaks in the United States were caused by fresh produce [11]. In 2011, one of the largest outbreaks of a foodborne infection by Shiga-toxin-producing *Escherichia coli* O104:H4 occurred in Germany, which originated from contaminated seed sprouts [12,13]. It resulted in nearly 3000 cases, with more than 20% associated with hemolytic uremic syndrome (HUS). Aside from pathogenic bacteria, opportunistic pathogenic bacteria also pose a threat as they can cause serious infections among immunocompromised people, young children, and elderly people. Among such bacteria, *Klebsiella* spp. and *Enterobacter* spp. have been found on produce obtained from retail markets [10,14,15].

The persistent detection of harmful bacteria, including antibiotic-resistant, opportunistic pathogens are of particular concern, since such foods can introduce antibiotic-resistant bacteria and/or resistance genes (ARGs) into the human body and make bacterial infections difficult to treat [16]. Multidrug-resistant *E. coli* and *Salmonella* spp. in vegetables have also often been reported as causative agents of foodborne outbreaks in Europe and North America [17,18,19,20,21]. These cases demonstrate the importance of regularly monitoring antibiotic-resistant bacteria in agricultural products and controlling the usage of antibiotics in the agricultural environment.

Despite the continuous threats posed by antibiotic-resistant bacteria in fresh produce, their potential risks remain largely unexplored. Recently, the ARGs in ready-to-eat (RTE) foods from southern China were investigated and it was found that multidrug-resistant genes including chloramphenicol, aminoglycoside, tetracycline, and beta-lactam resistance were the predominant ARG types in RTE foods [22]. As evidenced by this research, there is a growing need to investigate fresh produce in the context of antibiotic resistance in Korea, especially as the use of animal manure in agriculture is particularly known for this country.

There have been a few attempts to investigate antibiotic-resistant bacteria from fresh produce in South Korea [23,24]. Moreover, these studies focused only on *E. coli* and *Klebsiella* (*K.*) *pneumoniae*, not on the whole bacterial population. Additionally, these studies relied on using cefotaxime as a selective antibiotic to find ESBL-producing strains, but did not consider resistance to other classes of antibiotics that may also play a pivotal role in the spread of antibiotic resistance genes [23,24]. Thus, an understanding of the predominant bacteria found on vegetables and in the relevant environments and their antibiotic resistance profiles are yet to be understood. This study, therefore, aimed to narrow this knowledge gap and further evaluate risks associated with antibiotic resistance in fresh produce in South Korea. Specifically, it aimed to identify and characterize bacteria isolated from lettuce cultivated in Korea by using phenotypic and genotypic methods. Among different phyla of bacteria, this research focused on *Proteobacteria* such as *Klebsiella*, *Acinetobacter*, *E. coli*, and *Pseudomonas*, as these may harbor antibiotic-resistant strains.

## 2. Materials and Methods

### 2.1. Sampling

Leafy lettuces, manure, and soil samples were obtained from test beds in the Korean National Institute of Agricultural Sciences (NAS) between 29 June 2021 and 28 July 2021. To determine the occurrence of antibiotic-resistant bacteria in lettuce leaves and surrounding soils, we collected samples from both these sources. Lettuce seedlings were grown for 30 days in the field using different fertilized soils. After 30 days, two areas of harvested lettuce leaves (two times approximately 40 g) and their surrounding soils (two times ca. 200 g) were collected. This study included six different fertilized soils: cow manure (i), pig manure (ii), poultry manure (iii), chemical fertilizer (iv), fertilizer with mixed cow and poultry manure (v), and fertilizer with mixed cow, pig, and poultry manure (vi), as well as soil without fertilizer (seven sample types in total) (Table 1). All 28 samples were immediately stored and transported to the NAS laboratory at 4 °C and microbiologically analyzed within 4 h.

### 2.2. Microbiological Analyses

Twenty-five grams of lettuce samples were collected from each of the 40 g samples, and 25 g of soil samples were collected from each of the 200 g surrounding soil samples from the seven different test beds. These samples were each placed in stomacher bag (3M, Seoul, Korea) with 225 mL of buffered peptone water (Difco, Sparks, MD, USA) (1:10 dilution) and homogenized in a stomacher (Interscience, Saint-Nom la Breteche Arpents, France) for 2 min at maximum speed. For the detection and enumeration of bile-tolerant Gram-negative bacteria, tenfold serial dilutions of the homogenized samples were prepared with quarter-strength Ringer’s solution (Merck, Darmstadt, Germany) and 0.1 mL samples of appropriate dilutions were spread-plated onto violet red bile dextrose agar (VRBD) (Merck) plates. After incubation for 24 h at 37 °C, well-separated colonies of different morphologies were randomly picked from each plate. A total of 248 bacterial isolates (five to ten colonies from each sample) were selected and transported to the Max Rubner-Institut in Germany at an ambient temperature by courier. These strains were then further purified by repeated streaking on Luria–Bertani (LB) agar (Roth, Karlsruhe, Germany) plates. Some strains had two different morphologies on LB agar plates, so they were considered as separate strains and further purified on LB and Brain Heart infusion (BHI) (Merck) plates. Purified strains were then grown in LB broth (Roth) and stored in LB medium containing 30% glycerol (Roth) (*w*/*v*) at −80 °C.

### 2.3. Preliminary Phenotypic Testing

The purpose of this study was to identify and characterize the 248 Gram-negative isolates from 28 different samples (Table 1). In order to get a presumptive identification of the bacteria, phenotypic tests were done first. The Gram reaction of all the strains was confirmed using the 3% KOH (*w*/*v*) method on LB agar plates after 24 h. In addition, catalase and oxidase enzyme activities were tested with a 3% H_2_O_2_ (Merck) (*v*/*v*) solution and with oxidase test strips (Roth), respectively, to distinguish members of the *Enterobacteriaceae* family [25].

### 2.4. Genotypic Characterization by RAPD-PCR Fingerprinting

Random amplified polymorphic DNA polymerase chain reaction (RAPD-PCR) fingerprinting was done to characterize the strains at the strain level and to exclude potential multiple isolates of the same strain. For genotyping the 248 strains, RAPD-PCR fingerprinting was performed [26]. Total genomic DNA of strains grown in LB broth overnight at 37 °C was extracted using the peqGOLD Bacterial DNA Kit (VWR, Darmstadt, Germany) and the final DNA concentration was measured using Nanodrop (Peqlab, Erlangen, Germany). The extracted DNA served as a template and the M13 primer (5′-GAG GGT GGC GGT TCT-3′) was used to amplify random fragments of DNA. In a 25 µL PCR reaction, the PCR amplification mixture contained 2 µL of template DNA, 1 µmol/mL of M13 primer, 300 µM of each dNTP (Roth), 1.5 U of Taq-Polymerase E (Genaxxon Bioscience, Ulm, Germany), and 1X PCR buffer (Genaxxon Bioscience). The PCR products were then amplified using an initial denaturation at 94 °C for 2 min, followed by 40 cycles of denaturation at 94 °C for 1 min, primer annealing at 45 °C for 20 s, and extension at 72 °C for 2 min, which was ramped at 0.3 °C/s. This was followed by a final extension step at 72 °C for 5 min. To confirm the PCR products, gel electrophoresis was performed in 1.5% (*w*/*v*) agarose gel at 100 V for 4 h (Bio-Rad, Feldkirchen, Germany) [27]. The gels were then stained for 1 h with GelRed (Merck) and de-stained for 20 min with deionized water. For RAPD cluster analysis, the BioNumerics Version 8.0 (Applied Maths, Sint-Martens-Latem, Belgium) software was used. Clustering analysis was carried out with means of the Pearson correlation coefficient and the unweighted pair group method with arithmetic mean (UPGMA). After RAPD-PCR genotyping, only 202 of the 248 strains were selected for further antibiotic resistance testing, as the RAPD-PCR fingerprints suggested that some strains (with highly similar fingerprint profiles of >80% similarity) were multiple isolates of the same strain (Appendix A).

### 2.5. Antibiotic Susceptibility Test

Antibiotic susceptibility was tested by the Kirby–Bauer disc diffusion method according to the Clinical Laboratory Standards Institute (CLSI). The antibiotic discs obtained from Oxoid (Wesel, Germany) used for susceptibility testing including ampicillin (AMP, 10 µg), cefotaxime (CTX, 30 µg), cefoxitin (FOX, 30 µg), meropenem (MEM, 10 µg), ciprofloxacin (CIP, 5 µg), chloramphenicol (C, 30 µg), gentamicin (CN, 10 µg), tobramycin (TOB, 10 µg), streptomycin (S, 10 µg), and tetracycline (TET, 30 µg). The inoculum size was ca. 1 × 10^5^ CFU/mL of overnight culture and inoculated Mueller–Hinton (MH) (Roth) plates were incubated at 35 °C for 18 h as described in the CLSI guideline [28]. After incubation, the diameter of the inhibition zone was measured, and the isolates were grouped into susceptible, intermediate, or resistant categories according to the CLSI criteria. All antibiotic resistance tests were performed in duplicate. Of the 202 strains initially selected by RAPD-PCR fingerprinting analysis (< 80% similarity in fingerprint profile), 10 strains did not grow in MH agar (Roth), so their antibiotic susceptibility could not be determined.

### 2.6. 16S rRNA Gene Sequencing and Identification

The 16S rRNA gene PCR and sequence analysis were performed to identify 202 strains selected based on the RAPD cluster analysis after disregarding 46 strains considered as multiple isolates of the same strain (Appendix A). For sequencing, both the Nanopore Flongle DNA sequencing and Sanger sequencing methods were used. Most strains (141 isolates out of 202; 69.8%) were sequenced with the Nanopore DNA sequencing method, as it enabled easy and quick long-read DNA sequencing. For this application, barcodes and spacer sequences were added to the modified a 27F (5′-AGR GTT TGA TCM TGG CTC AG-3′) and 1492R (5′-TAC CTT GTT ACG ACT T-3′) primer as described in a previous study [26]. However, for strains that failed to sequence by Nanopore Flongle DNA sequencing, the 16S rRNA gene was bidirectionally sequenced by commercial Sanger sequencing at Microsynth (Göttingen, Germany). In the PCR step, both methods used the same genomic DNA as a template which was previously isolated for RAPD. However, each method required different primers. For Nanopore sequencing, 96 sets of forward and reverse primers which were designed in house were used [26]. For Sanger sequencing, the 16S rRNA gene was amplified using the forward primer 27F (5′-AGA GTT TGA TCM TGG CTC AG-3′) and the reverse primer 1540R (5′-TAC GGY TAC CTT GTT ACG ACT-3′). The PCR for both the Nanopore Flongle sequencing and Sanger sequencing contained 5 µL of template DNA (ca. 100 ng DNA), 500 pmol/mL of each primer, 150 µM of each dNTP (Roth), 1.5 U of Taq-Polymerase E (Genaxxon), and 1× PCR buffer (Genaxxon) in a 50 µL volume. The PCR product was then amplified using initial denaturation at 94 °C for 3 min, followed by 32 cycles of denaturation at 94 °C for 30 s, primer annealing at 55 °C for 30 s, and extension at 72 °C for 1 min 30 s. This was followed by a final extension step at 72 °C for 5 min. To confirm the PCR product, gel electrophoresis was performed using a 1.5% (*w*/*v*) agarose gel stained with 0.1 U of GelRed (Merck) at 100 V for 1 h. For Nanopore Flongle sequencing, the DNA pool with 16S PCR products was prepared and loaded onto the flow cell according to the protocol provided by Oxford Nanopore Technologies [29]. For Sanger sequencing, the PCR product was cleaned using a NucleoSpin gel and PCR clean-up kit (Macherey-Nagel, Düren, Germany), sent for commercial sequencing at Microsynth Seqlab (Göttingen, Germany), and identified by the EzBioCloud database (http://www/ezbiolcloud.net/identify (Database v. 07 July 2021, accessed on 22 August 2022)). Nanopore Flongle sequencing data were analyzed with the NanoCLUST [30] pipeline and each consensus sequence belonging to a specific strain was identified with the EzBioCloud platform (http://www/ezbiolcloud.net/identify (Datavase v. 07 July 2021, accessed on 22 August 2022)).

### 2.7. Whole Genome Sequencing

In this study, the whole genomes of fifteen randomly chosen isolates from each of the genera identified in this study (including isolates from the top four frequently isolated genera *Pseudomonas*, *Acinetobacter*, *Pantoea*, and *Enterobacter*, as well as, *Serratia*, *Lelliottia*, *Erwinia*, *Klebsiella*, and *Aeromonas*) were sequenced in order to investigate the genomic characteristics. The fifteen selected isolates were grown overnight in LB broth at 37 °C and the genomic DNA was extracted for sequencing using the peqGOLD Bacterial DNA Kit (VWR). The concentration and quality of extracted DNA were measured using a NanoDrop (Peqlab, Erlangen, Germany) and a Qubit 3 spectrophotometer (Invitrogen, Darmstadt, Germany), respectively. The TruSeq Nano DNA library preparation kit was used. The NextSeq reagent kits were used for the 2 × 150 bp paired-end sequencing with NextSeq 500 according to the manufacturers’ instructions (Illumina, Munich, Germany). Isolates were sequenced with the Nextseq 500 sequencing platform. The raw sequence data were trimmed using Trimmomatic pipeline (v. 0.32; parameters: Phred 33, sliding window; 4:15, leading; 3, and minlen; 45) [31] and de novo assembly was subsequently performed using SPAdes pipeline (v. 3.15.5; parameters: --isolate) [32]. After genome assembly, contig sequences which were shorter than 500 bp or contaminated with the spiked PhiX genome sequence were removed using the BBDuk pipeline (BBDuk Guide—DOE Joint Genome Institute). All post-processed contigs were annotated by prokka (v. 1.12) with the default parameter [33] and the NCBI Prokaryotic Genome Annotation Pipeline (v. 4.13).

PlasmidFinder was used to detect plasmid-related sequences, while ResFinder was used to detect acquired and/or point mutated antibiotic resistance genes. These analyses were performed using the staramr pipeline (v. 0.7.0) with default parameters [34]. The databases used in the staramr analyses were the PlasmidFinder [35] and ResFinder [36] databases, respectively. In order to precisely identify whole-genome-sequenced isolates, the average nucleotide identity was calculated using the OrthoANI pipeline [37] (v. 1.2) with the USEARCH tool and the digital DNA–DNA hybridization (dDDH) values were calculated using the Genome-to-Genome Distance Calculator (http://ggdc.dsmz.de/ggdc.php# (v.3.0, accessed on 23 January 2023)), applying formula 2 [38].

## 3. Results

### 3.1. Preliminary Physiological Tests

Of the 248 strains isolated, 246 isolates (99.2%) were Gram-negative bacteria based on the KOH Gram-determining method. Only two isolates (0.8%) were Gram-variable as they could not be accurately determined to be either KOH-positive or negative based on the KOH test. Since none of the strains were Gram-positive, further physiological tests such as oxidase and catalase tests were performed for all 248 strains. In total, 183 isolates (73.8%) were oxidase-negative. Strains showing an oxidase-negative reaction were considered indicative of *Enterobacteriaceae*, while oxidase-positive bacteria indicated *Pseudomonadaceae*. A total of 93.8% were catalase-positive, which is generally reported for *Enterobacteriaceae*. In this study, most of our isolates were Gram-negative, oxidase-negative, and catalase-positive strains which were presumptive *Enterobacteriaceae* strains.

### 3.2. RAPD-PCR Fingerprinting

The RAPD cluster analysis (Appendix A) was based on Pearson correlation and UPGMA parameters together with 2% band optimization. In this study, strains with highly similar band patterns that displayed similarity values equal to or higher than ca. 80% in RAPD clustering were considered as multiple isolates of the same strain. Forty-six isolates were grouped together with more than one isolate over 80% similarity (not underlined in Appendix A) and were thus excluded for the further molecular study. A total of 202 single isolates were further investigated for antibiotic resistance in this study (underlined in Appendix A).

### 3.3. Strain Identification

In this study, 202 strains from the 28 lettuce samples cultivated in South Korea and the surrounding soils were selected as distinct strains. The principal method of 16S rRNA gene sequencing used in this study was Nanopore Flongle sequencing. However, as 66 strains could not be sequenced either because of contamination of sequencing inhibitors or lack of DNA, those strains were sequenced by Sanger sequencing. From these 202 strains, 184 could be assigned either to the genera *Pseudomonas* (*n* = 61, 30.2%), *Acinetobacter* (*n* = 29, 14.4%), *Pantoea* (*n* = 27, 13.4%), *Enterobacter* (*n* = 19, 9.4%), *Flavobacterium* (*n* = 14, 6.9%), *Lelliottia* (*n* = 10, 4.9%), *Serratia* (*n* = 9, 4.5%), *Erwinia* (*n* = 4, 1.9%), *Aeromonas* (*n* = 3, 1.5%), *Brucella* (*n* = 3, 1.5%), *Klebsiella* (*n* = 3, 1.5%), *Leclercia* (*n* = 1, 0.5%), and *Pluralibacter* (*n* = 1, 0.5%) based on 16S rRNA gene sequencing using the EzTaxon database for identification [39] (Figure 1). However, eighteen isolates (8.9%) could not be unequivocally identified to the genus level by the 16S rRNA gene sequencing method as the sequencing result identified two or more related genera (Figure 1).

### 3.4. Antibiotic Resistance Profiles

All 202 strains were screened for antibiotic resistance using the disc diffusion method according to the CLSI guidelines. Nine *Pseudomonas* spp. and one *Acinetobacter* spp. out of the two hundred and two strains did not grow on MH agar plates and could therefore not be used for antibiotic susceptibility testing. The results classified strains as being either susceptible, intermediate, or resistant based on the diameter of the inhibition zone size. In this study, 133 (69.3%) and 105 (54.7%) strains showed a resistance phenotype to ampicillin and cefoxitin, respectively (Figure 2). Of the 192 strains tested in this study, 23 (11.9%) strains were fully susceptible to the antibiotics, while 30 (15.6%) strains showed an intermediate phenotype and were resistant to at least one antibiotic. A total of 84 strains were classified as susceptible to cefotaxime, while 82 (42.7%) strains and 26 (13.5%) strains showed resistant and intermediate phenotypes, respectively. For each antibiotic, strains belonging to the *Pseudomonadaceae* showed a generally high level of resistance at the family level. The resistance phenotype to gentamicin, tobramycin, ciprofloxacin, and tetracycline occurred only at low incidences of 1.6%, 1.6%, 1.0%, and 0.5%, respectively. However, a few isolates exhibited an intermediate phenotype to ciprofloxacin (*n* = 21, 10.9%), tobramycin (*n* = 6, 3.1%), and tetracycline (*n* = 7, 3.6%) (Figure 2). Only eight strains (six *Pseudomonas* spp., one *Klebsiella* spp., and one *Acinetobacter* spp.) were resistant to meropenem, which is a last-resort antibiotic for multidrug-resistant bacterial infections.

### 3.5. Whole Genome Sequencing

The genomes of fifteen selected isolates were sequenced and de novo assembled to generate draft genome sequences. The total raw sequence data obtained for genome coverage of these strains ranged from 25 (strain V98_8) to 147 (strain V89_4). The contig numbers of the fifteen isolates after sequencing ranged between 21 and 113, and the genome sizes ranged between 3.27 and 6.51 Mbp (Table 2). The highest G+C content (mol%) was 62.3% for strain V104_10 and the lowest was 38.64% for strain V89_4. The N_50_ values ranged from 115019 (V98_8) to 2533332 (V114_1) and the number of CDS were between 3116 and 6043. Six of these strains (V98_8, V88_4, V104_6, V104_10, V89_7, and V90_4) did not possess any sequences related to acquired antibiotic resistance, whereas nine strains contained more than one sequence related to potentially acquired antibiotic resistance from the ResFinder database. Of all the isolates which were fully genome-sequenced in this study, only the isolate V90_4 was found to contain an *IncFII* sequence related to a Gram-negative plasmid incompatibility type, by using the PlasmidFinder database (Table 2). Plasmids belonging to this incompatibility type are known to often be transferable and to carry antibiotic resistance [40]. In addition, the ResFinder database detected acquired antibiotic resistance genes in the other genome-sequenced strains, including a gene encoding a fosfomycin resistance protein *fosA* and *fosA2*, genes for the antibiotic efflux pump *OqxA* and *QqxB*, (fluoro)quinolone-resistance gene *qnrE1,* and β-lactamase genes *cphA1*, *ampH*, *blaMIR-6*, blaMOX-4, *blaACT-12*, *blaOXA-304*, *blaADC-25*, *blaSST1*, and *blaOKP-A-11*. Furthermore, genes for aminoglycoside resistance (*aac*(6′)-*Ic*) and for tetracycline resistance (*tet*(*41*)) could also be detected among the sequenced strains.

All of the genome-sequenced isolates were precisely identified using the average nucleotide identity (ANI) calculations and digital DNA–DNA hybridization values (dDDH) following the minimum guideline for species identification by these methods proposed in a previous study (Table 2) [41]. Except for the V89_13 strain, all strains that had generated draft genome sequences in this study were clearly identified by ANI and DDH analysis, along with closely related type strains above the species delineation cutoff values, and the results of the identification are shown in Table 2. Interestingly, the isolate V 89_13 showed the highest relationship to the *Lelliottia jeotgali* PFL01^T^ type strain, with an ANI value of 91.12% similarity. Additionally, the dDDH value between V89_13 and the closest type strain *Lelliottia jeotgali* PFL01^T^ was 43.6% (Table 2). This showed that both the ANI and in silico DDH analysis values for strain V89_13 and the closest type strain *Lelliottia jeotgali* PFL01^T^ were below the cutoff value used for species delineation (Table 2), indicating that the strain probably represented a novel *Lelliottia* species, which will be further taxonomically investigated in follow-up studies.

**Table 2 microorganisms-11-01241-t002:** Summary of whole genome sequencing of 15 selected isolates. n.d., not detected; AMP, ampicillin; C, chloramphenicol; S, streptomycin; CIP, ciprofloxacin; MEM, meropenem; CTX, cefotaxime; FOX, cefoxitin.

	V98_8	V88_4	V104_6	V104_10	V89_4	V89_7	V106_11	V108_6	V87_3
No. of contigs	113	48	58	66	30	34	37	58	35
N_50_	115,019	382,325	138,943	195,255	250,587	215,337	320,651	163,040	211,224
GC content (mol%)	59.99	60.55	61.77	62.3	38.64	43.03	53.33	53.37	55.70
Total length (bp)	6,514,074	6,264,170	4,687,376	5,620,582	4,009,586	3,276,090	4,899,709	4,944,362	4,706,154
Genome coverage	x 25	x 103	x 46	x 32	x 147	x 68	x 41	x 63	x 104
No. of CDSs	6043	5725	4326	5239	3771	3116	4956	5032	4443
No. of tRNAs	49	53	62	56	45	65	57	40	47
No. of rRNAs	3	5	6	5	3	3	5	5	4
Acquired resistance gene(s)	n.d.	n.d.	n.d.	n.d.	*bla*OXA-304, *bla*ADC-25	n.d.	*OqxB*	*OqxB*	*fosA*, *bla*MIR-6
Plasmid sequence(s)	n.d.	n.d.	n.d.	n.d.	n.d.	n.d.	n.d.	n.d.	n.d.
Antibiotic resistance	n.d.	C, AMP, FOX, CTX	n.d.	AMP, FOX, CTX	C, AMP, FOX, CTX	AMP, MEM, CTX	n.d.	AMP	AMP, FOX
OrthoANI identification (% similarity of top-hit)	*Pseudomonas umsongensis* DSM 16611^T^ (96.75%)	*Pseudomonas glycinae* MS586^T^ (96.48%)	*Pseudomonas fulva* DSM 17717^T^ (99.48%)	*Pseudomonas monteilii* DSM 14164^T^ (98.0%)	*Acineotbacter oleivorans* JCM 16667^T^ (96.91%)	*Acinetobacter soli* KCTC 22184^T^ (98.53%)	*Pantoea ananatis* LMG 2665^T^ (99.07%)	*Pantoea ananatis* LMG 2665^T^ (99.16%)	*Enterobacter cancerogenus* ATCC 33241^T^ (98.55%)
in silico DDH identification (% similarity of top-hit)	*Pseudomonas umsongensis* DSM 16611^T^ (71.9%)	*Pseudomonas glycinae* MS586^T^ (71.1%)	*Pseudomonas fulva* DSM 17717^T^ (96.1%)	*Pseudomonas monteilii* DSM 14164^T^ (83.4%)	*Acineotbacter oleivorans* JCM 16667^T^ (72.9%)	*Acinetobacter soli* KCTC 22184^T^ (88.2%)	*Pantoea ananatis* LMG 2665^T^ (92.5%)	*Pantoea ananatis* LMG 2665^T^ (93.2%)	*Enterobacter cancerogenus* ATCC 33241^T^ (87.4%)
Accession no.	JASCAE000000000	JASCAF000000000	JASCAG000000000	JASCAH000000000	JASCAI000000000	JASCAJ000000000	JASCAK000000000	JASCAL000000000	JASCAM000000000
	V87_3	V89_11	V114_1		V89_13	V90_4	V115_8		V90_14
No. of contigs	35	58	21		46	79	52		49
N_50_	211,224	201,295	2,533,332		173,726	134,120	194,518		197,099
GC content (mol%)	55.70	54.60	59.89		55.84	56.41	58.21		61.53
Total length (bp)	4,706,154	4,700,025	4,932,611		4,846,069	5,149,412	5,253,824		4,749,641
Genome coverage	x 104	x 115	x 41		x 85	x 103	x 40		x 127
No. of CDSs	4443	4509	4720		4561	4976	5084		4422
No. of tRNAs	47	36	76		48	52	65		56
No. of rRNAs	4	4	4		5	6	6		5
Acquired resistance gene(s)	*fosA*, *bla*MIR-6	*bla*ACT-12, *OqxA*, *OqxB*, *fosA*2	*aac*(6′)-*Ic*, *bla*SST-1, *OqxB*, *tet*(41)		*fosA*2, *OqxA*, *OqxB*	n.d.	*OqxA*, *OqxB*, *fosA*, *bla*OKP-A-11		*ampH*, *bla*MOX-4, *cphA*1
Plasmid sequence(s)	n.d.	n.d.	n.d.		n.d.	IncFII(Yp)	n.d.		n.d.
Antibiotic resistance	AMP, FOX	AMP, FOX	AMP		n.d.	AMP	CIP, AMP, MEM, CTX		AMP
OrthoANI identification (% similarity of top-hit)	*Enterobacter cancerogenus* ATCC 33241^T^ (98.55%)	*Enterobacter ludwigii* DSM 16688^T^ (98.88%)	*Serratia marcescens* ATCC 13880^T^ (98.64%)		*Lelliottia jeotgali* PFL01^T^ (91.19%), *Lelliottia amnigena* LMG2784^T^ (85.24%), *Lelliottia nimipressuralis* CIP 104980^T^ (84.15%)	*Erwinia aphidicola* JCM 21238^T^ (99.02%)	*Klebsiella quasipneumoniae* subsp. *quasipneumoniae* 01A030^T^ (99.19%)		*Aeromonas hydrophila* ATCC 7966^T^ (96.91%)
in silico DDH identification (% similarity of top-hit)	*Enterobacter cancerogenus* ATCC 33241^T^ (87.4%)	*Enterobacter ludwigii* DSM 16688^T^ (91.4%)	*Serratia marcescens* ATCC 13880^T^ *(*89.1%)		*Lelliottia jeotgali* PFL01^T^ (43.6%), *Lelliottia nimipressuralis* CCUG 25894^T^ (27.8%), *Lelliottia amnigena* LMG2784^T^ (29.1%)	*Erwinia aphidicola* JCM 21238^T^ (91.9%)	*Klebsiella quasipneumoniae* subsp. *quasipneumoniae* 01A030^T^ (93.9%)		*Aeromonas hydrophila* ATCC 7966^T^ (73.3%)
Accession no.	JASCAM000000000	JASCAN000000000	JASCAO000000000		JASCAP000000000	JASCAQ000000000	JASCAR000000000		JASCAS000000000

## 4. Discussion

In this study, Gram-negative bacteria isolated from lettuce and the surrounding soil in Korea were identified and characterized to obtain a preliminary understanding of the safety of these types of produce. The microbial ecology of agricultural environments was investigated using a culture-based method, focusing on a particular microorganism, i.e., *Enterobacterales*, by using selective media for these bacteria. RAPD-PCR was successfully used to discriminate distinct strains among multiple isolates of the same strain. In addition to the RAPD fingerprint comparison, 16S rRNA sequencing was applied to identify all isolates using the EZTaxon database. After both RAPD comparison and 16S-sequence-based identification, 202 strains of out 248 isolates were selected and the predominating families of the 202 strains were *Pseudomonadaceae* (30.2%), *Enterobacteriaceae* (16.8%), *Erwiniaceae* (15.3%), and *Moraxellaceae* (14.4%). At the genus level, *Pseudomonas* (30.2%) was the most prevalent genus, followed by *Acinetobacter* (14.4%), *Pantoea* (13.4%), and *Enterobacter* (9.4%). A previous study showed that *Pseudomonas* was the predominant Gram-negative bacteria present in the processing of endive lettuce and the low-temperature storage of fresh-cut lettuce [2]. In another study that investigated lettuce samples in Korea using culture-independent 16S rRNA gene amplicon metagenome analysis, Gram-positive *Bacillus* and *Exiguobacterium*, and Gram-negative *Pseudomonas* were the dominant genera observed in lettuce [5]. However, it has been reported that the microflora compositions in lettuce can differ depending on the site and time of sample collection [5].

*Pseudomonas aeruginosa*, *Acinetobacter baumannii*, and *Enterobacter cloacae* are specific species within these genera *Pseudomonas*, *Acinetobacter*, and *Enterobacter*, respectively, and are considered typical opportunistic pathogens that can cause serious infections in infants, the elderly, and immunocompromised hosts. Furthermore, though detected in lower numbers (4.5%), *Serratia* is important since *Serratia marcescens*, which amounted to 55.6% of the total *Serratia* strains found, has recently emerged as an important cause of nosocomial infections [42]. Therefore, it is necessary to regularly monitor the prevalence of potential clinical pathogens belonging to Serratia, along with those included in the genera *Pseudomonas*, *Acinetobacter*, and *Enterobacter*.

In a previous metagenomic study, *Acinetobacter*, *Pantoea*, and *Enterobacter*, which were the dominant isolates from both lettuce and cultivating environments in our study, did not predominate in the Chinese cabbage or the cabbage farming environments in Korea [43]. Such a discrepancy might arise from differences in the microbiota between different products such as cabbage and lettuce, as well as differences in analysis methods between culture-independent metagenome analysis and culturing Gram-negative bacteria using VRBD agar plates. It can be advantageous to use the culture-independent method because it provides information about the total microbial community. However, in our study, culturing allowed for the detection of the subset of antibiotic-resistant, potentially pathogenic or pathogenic Gram-negative bacteria which may carry transferable antibiotic resistance genes. This, coupled with whole genome sequence (WGS) analysis of representative species applied together with several bioinformatic pipelines, was able to support rapid potential pathogen identification and the detection of acquired or intrinsic antibiotic resistance genes [44].

This study used Nanopore sequencing as a method for 16S rRNA gene sequencing. Applying this method for bacterial species identification is a novel approach, while being low cost and robust. By not only supporting the almost complete 16S rRNA gene but also controlling sequencing quality using the NanoCLUST [30] program, which generates a consensus sequence from long-read sequencing results of individual 16S rRNA PCR products of strains, it is possible to correct the error rate of the long-read sequencing method and to verify the purity of the 16S rRNA gene products. If PCR products are contaminated, different clusters are drawn on the plot based on calculations made by the NanoCLUST, which helps to confirm if the extracted DNA is pure or not, i.e., whether it stemmed from a pure culture or not [29].

Antibiotic resistance tested by the disc diffusion method revealed that the isolates from lettuce and the surrounding soil in the Korean agricultural environment tested were highly resistant to ampicillin (69.7%), with *Pseudomonas* (*P.*) being the dominant genus. The previous review study of Rahman et al. [16] reported that six of forty studies frequently isolated ampicillin-resistant *Pseudomonas* spp. as well as ESBL-producing *Pseudomonas* spp. from fresh produce samples. This ampicillin-resistant profile in the genus *Pseudomonas* is consistent with the study conducted with fresh produce, which found that *Pseudomonas* spp. Recovered from lettuce samples frequently exhibited resistance to ampicillin [45]. Additionally, culturable bacteria isolated from environmental water samples in Korea were found to be highly resistant to the ampicillin antibiotic [46]. In addition, the resistance to cefoxitin (54.7%), a second-generation cephalosporin antibiotic, was the second most frequently occurring resistance profile. This reflects the wide use of cefoxitin in human and veterinary medicines. A notable result was that 84 (42.7%) out of 192 isolates in this study also showed resistance towards cefotaxime. Of these strains, *Pseudomonadaceae* (54.8%) and *Moraxellaceae* (32.1%) were primarily identified. A previous study on *P. aeruginosa* by Pang et al. [47] showed the outer membrane permeability and efflux system of this strain caused a high level of intrinsic resistance to antibiotics. In this study, the *Pseudomonas glycinae* V88_4 isolate, for which non-acquired antibiotic resistance genes were detected in the ResFinder analysis, was resistant to ampicillin, cefoxitin, cefotaxime, and chloramphenicol. In addition, one strain identified as *Acinetobacter soli* V89_7 was also found to contain non-acquired antibiotic resistance genes by ResFinder, but showed resistance to ampicillin, meropenem, and cefotaxime. The high resistance to cefotaxime observed in these two strains could indicate that intrinsic resistance factors, such as point mutations in target genes, may be attributed to resistance.

Cefotaxime is a third-generation cephalosporin that is important for treating infections caused by members of the *Enterobacteriaceae*, i.e., *Enterobacter* spp., *Klebsiella* spp., and *Escherichia coli*. However, many clinical case studies reported isolates of these bacteria from infected patients to show resistance to this antibiotic [48]. Furthermore, the clinical *Enterobacteriaceae* isolates that were resistant to cefotaxime showed multidrug resistance properties. The primary reason for cefotaxime resistance in the study [48] was attributed to CTX-M-type beta lactamase genes, which were present in 76% of the *Enterobacteriaceae*, i.e., *E. coli* and *Klebsiella* spp. This implies that if cefotaxime-resistant enterobacterial isolates in fresh produce would possess ESBLs and that these would potentially be transferred to humans via horizontal gene transfer, this could result in a serious public health threat. In this study, it is possible that 42.7% of cefotaxime-resistant isolates are capable of producing one or more CTX-M-type beta lactamases. Therefore, further investigation is necessary to determine the presence of ESBL genes in these isolates using both physiological tests, i.e., MAST disc diffusion test, and molecular methods such as whole genome sequencing.

The incidence of chloramphenicol resistance among the bacteria in this study was 17.7%. Although chloramphenicol is not prescribed for humans and food-producing animals in the European Union (EU), it is still important to monitor the presence of chloramphenicol resistance genes in bacteria that are isolated from soil, plants, vegetables, and agricultural environments to ensure food safety for consumers [49]. Since chloramphenicol-resistant bacteria often display resistance to other antibiotics such as tetracycline and kanamycin, and harbor mobile genetic elements containing resistance plasmids and/or transposons [19].

Additionally, there was a low level of resistance towards aminoglycosides, which include gentamicin (1.6%), tobramycin (1.6%), and streptomycin (9.4%). According to the report published by the National Institute of Food and Drug Safety Evaluation and Ministry of Agriculture, Food and Rural Affairs, sales of streptomycin to livestock and fisheries in Korea were about 3.3 times higher than gentamicin, which might explain higher resistances observed in our experiments (www.mfds.go.kr (accessed on 22 August 2022)). A previous study of Schwaiger et al. [50] reported antibiotic-resistant *Pseudomonas* spp. isolated from vegetables at the marketing stage and showed similar resistance patterns as observed also in our study with regard to resistance incidences to gentamicin and tobramycin (<4%) and streptomycin (11%).

It is notable that there was low resistance to ciprofloxacin (1.0%) and to tetracycline (0.5%) in this study. Tetracyclines have in the past been used as therapeutic antibiotics in animal husbandry and for the control of plant diseases in Korea. The amounts of tetracyclines used in poultry flocks as therapy against diseases are still relatively high in several countries, which results in the accumulation of antibiotic substrates in the environment. This in turn, may influence the development of tetracycline resistance in foodborne bacteria, and the spread of these bacteria to the communities and hospitals via foods.

Carbapenem is another critical antibiotic that is used as a last resort to treat infections. From our experiment, 4.1% of bacteria were resistant to meropenem, which belongs to the class of carbapenem antibiotics. Primarily, *Pseudomonas* spp. (6), *Acinetobacter* spp. (1), and an *Enterobacteriaceae* (1) were found to be resistant to this antibiotic. Since genes related to resistance to carbapenem give further resistance to other antibiotics and often coexist with ESBL genes, it is highly likely that these resistant strains are potential ESBL-producing strains [7,16]. Furthermore, carbapenem-resistant, Gram-negative microorganisms have recently become critical issues in nosocomial infections. Among them, carbapenem-resistant *Acinetobacter baumannii*, *Pseudomonas aeruginosa*, *Escherichia coli,* and *Klebsiella pneumoniae* are WHO-designated ‘priority pathogens’ posing serious threats to human health [15]. As shown by clinical sample-based studies, genes including *blaCTX-M*, *blaSHV*, *blaTEM*, *blaOXA*, and *blaNDM* have been frequently found in carbapenem-resistant strains.

The genome sequences of 15 isolates were used to precisely identify the isolates as well as to determine the presence of potentially acquired antibiotic resistance genes, as well as plasmid-related sequences. ResFinder analysis showed a *Serratia* spp. isolate (V114_1) possessed resistance genes to antibiotics belonging to four different classes. Two strains (*Pseudomonas glyciane* V88_4 and *Acinetobacter soli* V89_7) which did not possess any acquired antibiotic resistance genes in the ResFinder analysis, appeared to be cefotaxime-resistant, because these strains were able to grow and showed only a small diameter inhibition zone (<18 mm) in resistance tests with this antibiotic. These discrepancies between phenotypic and genotypic resistance profiles may be due to the possible presence of new AMR gene variations which have not previously been identified, or on the other resistance mechanisms, such as efflux pumps and point mutations. This indicates that the determination of antimicrobial resistance should not be demonstrated only by searching for a specific gene that confers resistance to the antibiotic, but this should also be correlated with phenotypic resistance tests.

Genomic characterization of the fifteen Gram-negative bacteria from Korean soil and lettuce samples showed the low level (one out of fifteen strains) presence of potentially transferable antibiotic resistance genes on the genomes, indicating that the transfer of antibiotic resistances by potential pathogenic enterobacteria via fresh produce in Korea is a low but likely possibility. Fresh produce should therefore be continuously monitored for the occurrence and sources of entry of bacteria carrying transferable antibiotic resistances in order to find ways to lower the presence of these bacteria in the fresh produce food chain.

## Figures and Tables

**Figure 1 microorganisms-11-01241-f001:**
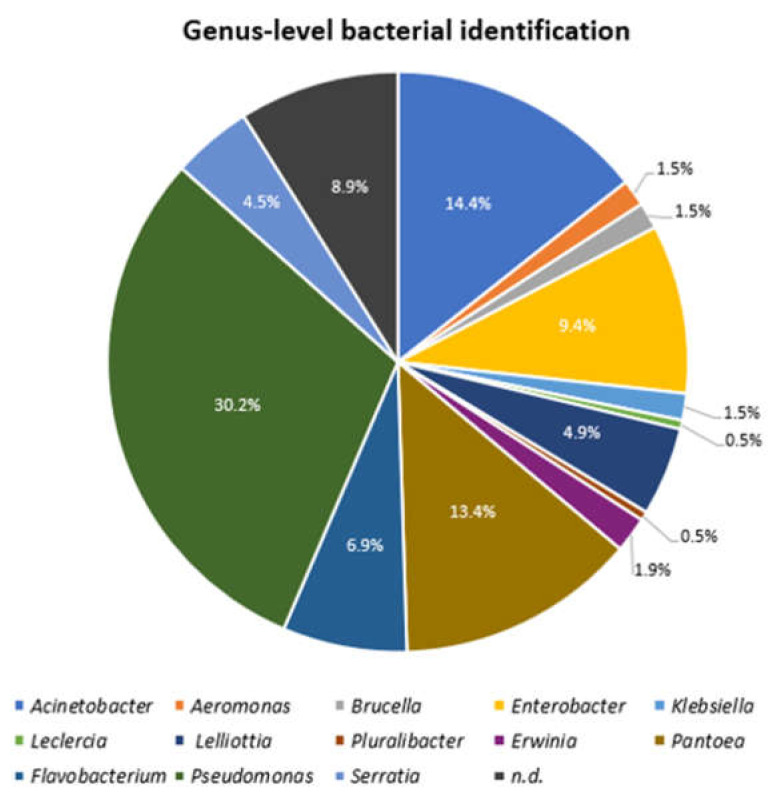
Bacterial identification of 202 Gram-negative strains using 16S rRNA gene sequencing based on the EzTaxon databases [39]. The percentages of each genera are indicated. n.d.; the sequencing result identified two or more closely related genera.

**Figure 2 microorganisms-11-01241-f002:**
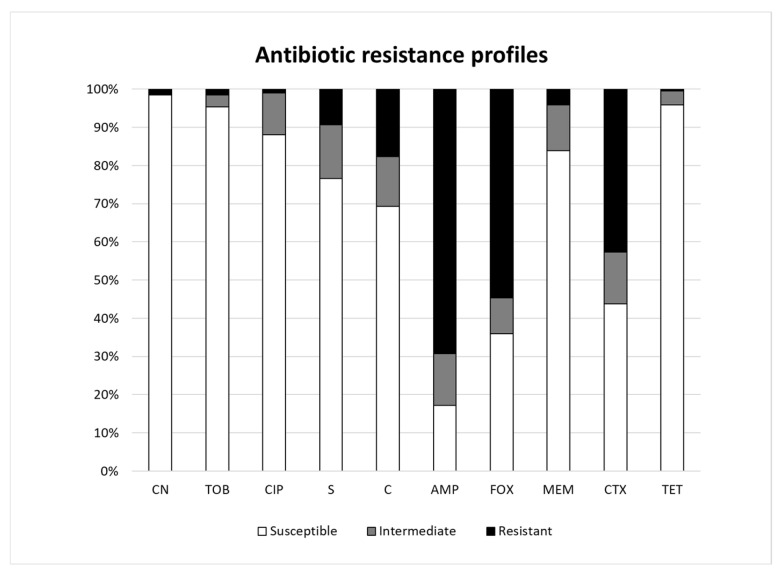
Antibiotic resistance profiles of Gram-negative bacteria isolated from lettuce and surrounding soils in Korea. Determination of antibiotic resistance based on inhibition zone diameters (mm) as resistant, susceptible, and intermediate according to the CLSI guidelines. CN, gentamicin; TOB, tobramycin; CIP, ciprofloxacin; S, streptomycin; C, chloramphenicol; AMP, ampicillin; FOX, cefoxitin; MEM, meropenem (MEM, 10 µg); CTX, cefotaxime; TET, tetracycline.

**Table 1 microorganisms-11-01241-t001:** Number of isolates obtained from the different types of surrounding soils and lettuces leaves.

Fertilizer Source	Surrounding Soils	Lettuce Leaves
Non-treated soil	20	44
Cow manure	5	26
Pig manure	3	25
Poultry manure	3	18
Chemical fertilizer	10	25
Chemical fertilizer + cow and poultry manure	12	23
Chemical fertilizer + cow + poultry + pig manure	13	21
Total number of isolates	66	182

## Data Availability

The draft genome sequences were deposited in DDB/ENA/GeneBank under the accession numbers listed in Table 2, PJ016303.

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
