# Peer review of "Identification and Characterization of Antibiotic-Resistant, Gram-Negative Bacteria Isolated from Korean Fresh Produce and Agricultural Environment"

_microorganisms, 2023, doi:10.3390/microorganisms11051241_

Round 1

Reviewer 1 Report

The manuscript aims to investigate antibiotic resistance issues in fresh produce in Korea. However there are several major concerns:

1. Table 2 is missing and I am assuming a lot of the information should be contained in Table 2

2. Sampling strategy is not representative. One produce, one location, one month of sampling is not representative regarding the entire Korean agriculture fresh produce industry. Certain claims need to be reworded and significant discussion need to be included to justify this sampling strategy.

3. Other comments are included in the attachment, include quite a few typos and sentences that need polishing.

Author Response

Thank you very much for your valuable comments.

Please find attached my responses. I have made changes to the manuscript based on your guidance, and I hope it now fits your expectations.

Reviewer 2 Report

This study is focused on the identification and characterization of antibiotic resistant and Gram-negative bacteria of Korean fresh products and surrounding soil. The manuscript is interesting and potentially suitable for Microorganisms journal, but some modifications should be carried out before publication. An English language and style editing are required.

ABSTRACT.

Line 17. Please add “that” after shown.

Line 18. Please replace “to be” with “could be”

INTRODUCTION.

Please control the name of the bacteria for all the manuscript. Some of them are not written in italics.

MATERIALS AND METHODS.

Please add the geographical location for some products (i.e. VWR, Roth, etc.)

Paragraph 2.1. Why did you choose these types of fertilized soils?

Paragraph 2.4. Is the method referred?

Paragraph 2.5. Line 167. Please correct the concentration.

Paragraph 2.6. Line 183. Please add “in a” before previous study.

DISCUSSION.

Line 354. Please remove “for this”.

Lines 365-367. Please explain better the sentence.

Lines 370. What are the genera which you referred?

Lines 395-398. Please rephrase the sentence.

Lines 424-426. Please rephrase the sentence.

Line 442. Please remove “this study” from the end of the sentence to the beginning of it.

Lines 446-449. Please rephrase the sentence.

Lines 502-504. What does it mean? Probably, you have to remove it before sending the manuscript.

Author Response

(The authors gave the same response as above.)

Round 2

Reviewer 1 Report

NA